# Proteins of generalist and specialist pathogens differ in their amino acid composition

Luz P Blanco[1],* , Bryan L Payne[2], Felix Feyertag[2], David Alvarez-Ponce[2]

Pathogens differ in their host specificities, with species infecting a unique host (specialist pathogens) and others having a wide host range (generalists). Molecular determinants of pathogen's host range remain poorly understood. Secreted proteins of generalist pathogens are expected to have a broader range of intermolecular interactions (i.e., higher promiscuity) compared with their specialist counterparts. We hypothesize that this increased promiscuity of generalist secretomes may be based on an elevated content of primitive amino acids and intrinsically disordered regions, as these features are known to increase protein flexibility and interactivity. Here, we measure the proportion of primitive amino acids and percentage of intrinsically disordered residues in secreted, membrane, and cytoplasmic proteins from pathogens with different host specificity. Supporting our prediction, there is a significant general enrichment for primitive amino acids and intrinsically disordered regions in proteins from generalists compared to specialists, particularly among secreted proteins in prokaryotes. Our findings support our hypothesis that secreted proteins' amino acid composition and disordered content influence the pathogens' host range.

## Introduction

Certain pathogens are highly species-specific, whereas others affect a broad range of hosts. However, the molecular bases of host range variability remain poorly understood (Pan et al, 2014). Secreted and membrane proteins are known to play important roles in host–pathogen interactions, and thus their interaction capabilities are expected to play a key role in determining host range, with generalist secreted and membrane proteins being capable of interacting with a wider range of molecules than their specialist counterparts.

Primitive amino acids (those believed to have predominated under prebiotic conditions, found in meteorite analysis, produced by the early genetic code, and which were spontaneously generated in Miller's spark discharge experiments [Doi et al, 2005; Parker et al, 2011]) have been associated with proteins with enhanced promiscuity and solubility. In contrast, non-primitive amino acids have been correlated with lower solubility, enhanced rigidity, and highly structured conformations (Davidson et al, 1995; James & Tawfik, 2003; Doi et al, 2005). Thus, we hypothesize that, for an organism to be highly promiscuous or capable of infecting/interacting with multiple hosts, their secreted and membrane proteins should be enriched in primitive amino acids compared with secreted and membrane proteins of host-restricted specialist pathogens. Primitive amino acids in surface exposed proteins might aid generalists' ability to interact with a broader range of proteins from different hosts. On the contrary, surface-exposed proteins from specialists do not require to interact with a high repertoire of proteins because they are host-restricted, and thus, they might have a reduced amount of primitive amino acids.

In addition, the protein's ability to establish intermolecular interactions is also dictated by their content of intrinsically disordered regions (Tompa et al, 2015; Ahrens et al, 2017). Intrinsically disordered regions are protein domains that lack a stable structural conformation and whose structure oscillates among different alternative conformations. Such regions confer flexibility to the proteins, facilitating their interactions with other molecules. Indeed, intrinsically disordered proteins (those with a high content of intrinsically disordered regions) tend to be highly connected (Dyson & Wright, 2002; Dunker et al, 2005). Thus, it is possible that the secreted proteins of the secreted and membrane proteins of generalist pathogens are highly disordered compared with those of specialists.

Here, we compare the frequency of primitive amino acid residues and the percentage of disordered regions of proteins from several pairs of generalists and specialists. In agreement with our predictions, we observe increased amounts of primitive amino acids and disordered regions predominantly in secreted proteins from generalists compared with specialist pathogens.

[1]Systemic Autoimmunity Branch, National Institute of Arthritis and Musculoskeletal and Skin Diseases, National Institutes of Health, Bethesda, MD, USA    [2]Department of Biology, University of Nevada, Reno, NV, USA

Correspondence: luz.blanco@nih.gov; dap@unr.edu

*Luz P Blanco contributed to this work in her personal capacity. The views expressed are her own and do not necessarily represent the view of the National Institutes of Health or the US Government.
Felix Feyertag's present address is Structural Genomics Consortium; Target Discovery Institute, Nuffield Department of Medicine, University of Oxford, Oxford, UK

**Table 1. Pathogens with their respective host range used in this study.**

| Host range | | Niche of infection | Type of pathogen | Reference or source |
|---|---|---|---|---|
| Specialist/host adapted (species) | Generalist | | | |
| *L. longbeachae* (*Homo sapiens*) | *L. pneumophila* | Facultative intracellular, lung environment | Bacteria, Gram negative | (Marra & Shuman, 1992; Whiley & Bentham, 2011)[a] |
| *S. enterica* Typhi (*H. sapiens*) | *S. enterica* Typhimurium | Facultative intracellular, systemic | Bacteria, Gram negative | (Sabbagh et al, 2010)[a] |
| *V. cholerae* O1 (*H. sapiens*) | *V. vulnificus* | Extracellular, mucosal pathogen | Bacteria, Gram negative | (Jones & Oliver, 2009; Safa et al, 2010)[a] |
| *M. leprae* (*H. sapiens*) | *M. bovis* | Intracellular, systemic and lung environment | Bacteria, actinobacteria | (Mikota & Maslow, 2011; Legendre et al, 2012)[a] |
| *B. quintana* (*H. sapiens*) | *B. henselae* | Facultative intracellular, vector transmissible, systemic | Bacteria, Gram negative | (Chomel et al, 2004; Veracx & Raoult, 2012)[a] |
| *B. recurrentis* (*H. sapiens*) | *B. burgdorferi* | Vector transmissible/extracellular, systemic | Bacteria, spirochete Systemic | (Veracx & Raoult, 2012)[a,b] |
| *C. pneumonia* (*H. sapiens*) | *C. trachomatis* | Obligate intracellular, lung environment/ sexually transmissible, eye infections | Bacteria | (Beare et al, 2011)[a] |
| *R. prowazekii* (*H. sapiens*) | *R. rickettsii* | Obligate intracellular, systemic | Bacteria, Gram negative | (Beare et al, 2011)[a] |
| *C. rodentium* (*Mus musculus*) | *C. freundii* | Extracellular, mucosal pathogen | Bacteria, Gram negative | (Borenshtein et al, 2008; Bai et al, 2012)[a,c] |
| *S. enterica* Gallinarum (*Gallus gallus*) | *S. enterica* Enteritidis | Facultative intracellular, systemic/ mucosal environment | Bacteria, Gram negative | (Braden, 2006; Blondel et al, 2013)[d] |
| *Streptococcus pyogenes* (*H. sapiens*) | *S. dysgalactiae* | Extracellular | Bacteria, Gram positive | (Cunningham, 2000; Abdelsalam et al, 2013)[a] |
| *Bacillus cereus* (human opportunistic) | *B. anthracis* | Extracellular/obligate intracellular | Bacteria, Gram positive | (Kotiranta et al, 2000; Spencer, 2003)[a] |
| *M. intracellulare* (*H. sapiens*) | *M. avium* | Intracellular, and systemic and lung environment | Bacteria, actinobacteria | (Claeys & Robinson, 2018)[a] |
| *C. albicans* (*H. sapiens*) | *C. glabrata* | Extracellular | Fungus | (Brunke & Hube, 2013; Mayer et al, 2013)[e] |
| *P. falciparum* (*H. sapiens*) | *B. microti* | Vector transmissible/extracellular and intracellular | Unicellular protozoa | (Gray et al, 2010; Prugnolle et al, 2011) |
| *W. bancrofti* (*H. sapiens*) | *A. suum* | Vector transmissible/egg consumption/ extracellular | Nematode | (Woolhouse et al, 2001; Nejsum et al, 2012) |

[a]https://my.absa.org/tiki-index.php?page=Riskgroups.
[b]http://www.phac-aspc.gc.ca/lab-bio/res/psds-ftss/msds21e-eng.php.
[c]https://www.msdsonline.com/resources/msds-resources/free-safety-data-sheet-index/citrobacter-spp/.
[d]http://www.sanger.ac.uk/resources/downloads/bacteria/salmonella.html.
[e]http://www.phac-aspc.gc.ca/lab-bio/res/psds-ftss/msds30e-eng.php.

# Results

The proteomes of 16 pairs of closely related pathogens, with each pair containing one generalist and one species-specific pathogen, were compared in this work (Table 1). These included 13 pairs of bacteria and 3 pairs of eukaryotes. Our list includes traditional human-specific bacterial pathogens such as *Salmonella typhi* (Sabbagh et al, 2010), *Vibrio cholerae* (Safa et al, 2010), and *Mycobacterium leprae* (Legendre et al, 2012), and other less well-characterized or recently emerged pathogens, such as *Legionella longbeachae* (Whiley & Bentham, 2011), *Bartonella quintana* (Veracx & Raoult, 2012), and *B. recurrentis* (Veracx & Raoult, 2012). These pathogens differ significantly in their transmission mechanisms and in their niches of infection. For comparison, broad generalists of the same species, the same genus, or closely related lineages, were chosen (Table 1). Eukaryote human-adapted pathogens included in our analyses were the fungus *Candida albicans* (Mayer et al, 2013), the unicellular protozoon *Plasmodium falciparum* (Woolhouse et al, 2001), and the nematode *Wuchereria bancrofti* (Woolhouse et al, 2001). For comparison, the fungus *C. glabrata* (Brunke & Hube, 2013), the unicellular protozoon *Babesia microti* (Gray et al, 2010), and the nematode *Ascaris suum* (Nejsum et al, 2012) were selected as broad host-range pathogens.

For each of the proteins of these organisms, we computed the ratio of primitive amino acids as AGVDE = [alanine (A) + glycine (G) + valine (V) + aspartic acid (D) + glutamic acid (E)]/(total amino acids).

This selection of primitive amino acids was based on Doi et al (2005). The ratio of primitive amino acids for proteins from prokaryote bacterial pathogens is significantly higher in generalist pathogens compared with specialist pathogens in all subcellular locations studied (secreted, membrane, and cytoplasmic; Fig 1A). For membrane proteins, the trend is significant for outer-membrane (extracellular) domains, trans-membrane domains, and inner-membrane (intracellular) domains (Fig 1A). The trend is particularly marked among secreted proteins and among the extracellular domains of membrane proteins. Similar trends are observed in eukaryotic pathogens; however, the difference is highly pronounced in all subcellular locations studied (Fig 1B). In addition to the AGVDE list, alternative lists of primitive amino acids have been proposed, including: (Ala, Gly, Asp, Val), (Gly, Ala, Glu, Val), and (Ala, Asp, Glu, Gly, Ile, Leu, Pro, Ser, Thr, Val) (Brooks et al, 2002; Doi et al, 2005; Longo et al, 2013). We thus repeated our analyses using these lists, with equivalent results (Tables S1 and S2).

To further expand and validate our observations, two pairs of bacterial obligate intracellular pathogens were compared: *Chlamydia pneumoniae* (human specialist) versus *C. trachomatis* (generalist) and *Rickettsia prowazekii* (human specialist) versus *R. rickettsii* (generalist) (Table 1). As shown in Tables S1 and S2, secreted proteins still show the enrichment in AGVDE, even though the trend is not so strong for membrane and cytoplasmic proteins. These observations suggest that primitive amino acid enrichment in bacterial secreted proteins is not due to differences in their niche of infection.

We next compared the proteomes of *S. enterica* Gallinarum (a *Gallus gallus* specialist pathogen) and *S. enterica* Enteritidis (a generalist pathogen). In agreement with our previous findings, secreted proteins from the generalist are enriched in primitive amino acids compared with the specialist; however, this bias is not present in membrane- or in cytoplasm-derived proteins (Tables S1 and S2). Therefore, the trend described here is not specific to human pathogens.

Primitive amino acids tend to require a small amount of energy for their synthesis (Li et al, 2009), thus raising the possibility that our observations might be due to generalists preferring economic amino acids rather than to amino acid primitiveness per se. To discriminate whether the trend described here in prokaryotes' secreted proteins is due to low energy cost or primitiveness, we studied the ratios of every amino acid separately. Supporting that amino acid primitiveness is the main contributor to the bias detected, glutamic acid in prokaryotes (average ratio in secreted proteins of generalist: 5.04%; average ratio in secreted proteins of specialist: 4.96%) and isoleucine in eukaryotes (average ratio in secreted proteins of generalist: 7.97%; average ratio in secreted proteins of specialists: 7.51%) are enriched in generalist secreted and membrane proteins, despite these being relatively expensive primitive amino acids for prokaryotes and eukaryotes, respectively (Fig S1) (Li et al, 2009). In addition, as expected, when we analyzed the ratio of expensive amino acids (RExAA) = (arginine [R] + histidine [H] + methionine [M] + phenylalanine [F] + tryptophan [W])/(total amino acids), trends were opposite to those for primitive amino acids: secreted proteins of host-specific bacteria exhibit a slightly higher RExAA than secreted proteins of generalist bacteria. However, differences were not statistically significant (Fig 1C). This

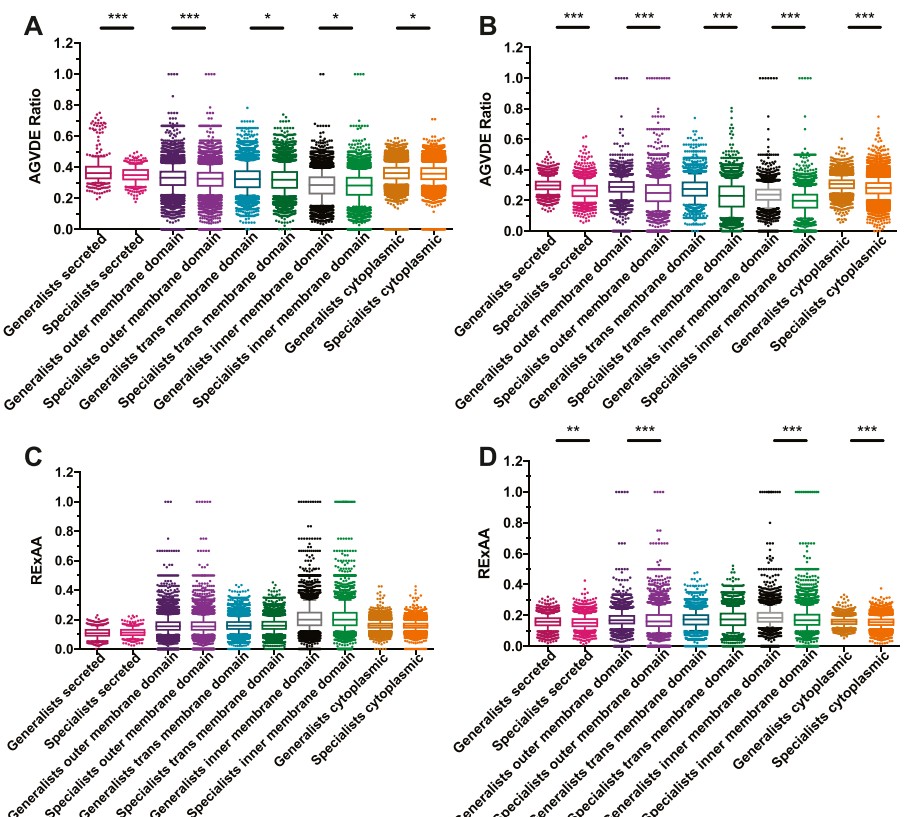

**Figure 1. Enrichment in primitive amino acids in secreted proteins from generalists.**
AGVDE ratio in generalists and specialists was determined in (A) for 13 pairs of prokaryotic pathogens and (B) for 3 pairs of eukaryotic pathogens, either in secreted, located in outer-, trans-, and inner-membrane stretches, or cytoplasm-derived proteins as indicated in Table 1. Similarly, in (C) and (D), the RExAA was determined. Shown are box-whiskers plots in which the line in the middle of the box represents the median value and whiskers are drawn down to the 10th and up to the 90th percentile. The *P*-values were determined using the Mann–Whitney *U* test. *P < 0.05; **P < 0.01; ***P < 0.001.

observation is probably because of the enrichment in primitive amino acids of secreted proteins from generalist bacteria in detriment of expensive amino acids. Nonetheless, the RExAA ratios of eukaryotic pathogens were significantly higher in generalists for almost all subcellular locations, particularly for inner-membrane domains of membrane proteins (Fig 1D).

Because proteins' intrinsic disorder is correlated with their ability to establish protein–protein interactions (Afek et al, 2011; Tompa et al, 2015), the percentage of disordered residues (both in short or long stretches) was quantified for all the proteins from the pathogens listed in Table 1. As shown in Fig 2A, the percentage of disordered residues for prokaryotic proteins is significantly higher in generalists compared with specialists. This bias is clearly stronger when considering long stretches of disordered residues and among secreted proteins compared with either membrane or cytoplasmic proteins (Fig 2A). A similar, but not as strong, bias in the percentage of disordered residues in secreted proteins was detected in eukaryotic pathogens (Fig 2B). However, short stretches of disordered residues are more prominently present in all subcellular locations and even predominate in the eukaryote specialists' cytoplasmic proteins compared with those of generalists (Fig 2B).

Significant correlations were observed between the percent of disordered residues and the AGVDE ratios, as shown in Fig 3A–H, particularly for secreted proteins of generalists and long-disordered stretches compared with specialists in prokaryote pathogens (Fig 3A versus C). In contrast, eukaryote pathogens do not follow the same patterns as prokaryotes pathogens (Fig 3C and G). Taking into consideration the prokaryotes' data, our hypothesis that secreted proteins from generalist pathogens should be enriched in disordered structures compared with specialist is supported.

The enrichment of generalists' secreted proteins in intrinsically disordered regions (Fig 2), combined with the positive correlation between intrinsically disordered content and primitive amino acid content (Fig 3), raises the possibility that the enrichment of generalists' secretomes in primitive amino acids may be simply a byproduct of their high degree of intrinsic disorder. Likewise, we found that the genes encoding generalists' secreted proteins exhibited a significantly higher guanine-cytosine (GC) content (Table S3), and that in some species GC content correlates with the content of primitive amino acids (Table S4), also potentially accounting for the enrichment of generalists' secretomes in primitive amino acids. To discard this possibility, we performed analysis of covariance analyses using the AGVDE ratio as the dependent variable, host range as a factor variable (generalist versus specialist), and intrinsic disorder and GC content at third codon positions (GC3) as explanatory variables. These analyses show that host range is a significant independent determinant of the content of primitive amino acids (Tables S5 and S6).

## Discussion

We describe an enrichment of the secreted proteins of generalist pathogens in primitive amino acids and in intrinsically disordered regions. These features are known to increase protein flexibility and interactivity (Tompa et al, 2015; Ahrens et al, 2017), which may help

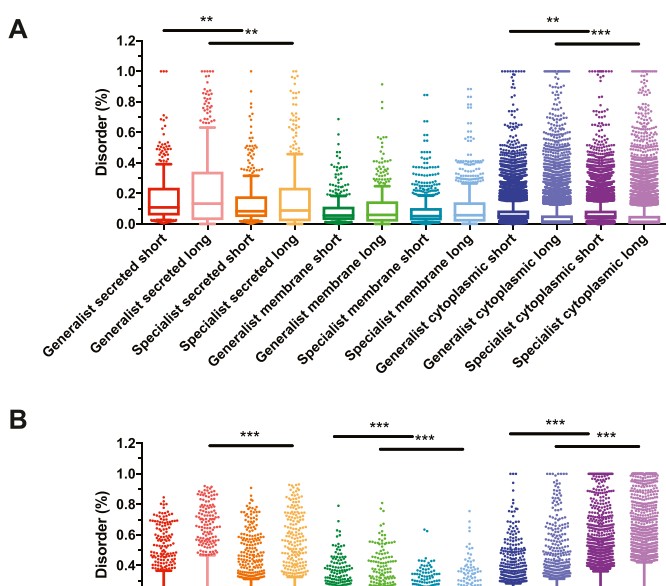

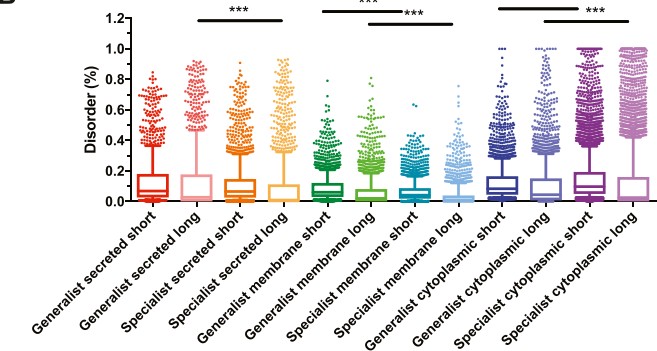

**Figure 2.  Enhanced intrinsic disorder in secreted proteins from generalists.**
The percentage of short or long stretches of disordered residues was determined in generalist and specialist pathogens in (A) for 13 pairs of prokaryotic pathogens and (B) for 3 pairs of eukaryotic pathogens, either in secreted, membrane, or cytoplasm derived proteins as indicated in Table 1. Shown are box-whiskers plots in which the line in the middle of the box represents the median value and whiskers are drawn down to the 10th and up to the 90th percentile. The P-values were determined using the Mann–Whitney U test. **P < 0.01; ***P < 0.001.

generalist pathogens to interact with a broader range of hosts. These findings support the proposition that bacterial secreted proteins are important in influencing the host range and that primitive amino acids predominate in secreted proteins of generalist pathogens.

An example of how the environment can bias the amino acid usage in proteins is the remarkable difference between the proteome composition of thermophilic versus mesophilic prokaryotes (Farias et al, 2004; Wang & Lercher, 2010a,b). In thermophiles, proteins are enriched in amino acids that contribute to thermostability such as arginine (R) (Linden & Farias, 2006). Moreover, the ratio (glutamic acid [E] + lysine [K])/(glutamine [Q] + histidine [H]) is elevated in thermophiles compared with mesophiles (Farias et al, 2004). In addition, proteins from thermophilic organisms are depleted in aspartic acid (D), asparagine (N), glutamine (Q), threonine (T), serine (S), histidine (H), and alanine (A) compared with proteins of mesophilic organisms (Taylor & Vaisman, 2010; Wang & Lercher, 2010a) and have reduced intrinsic disorder (Cherry, 2010). However, these biases are not dependent on protein location, in contrast to the biases described here, that is, a bias specific to secreted proteins. In addition, the generalist–specialist pairs included in our analyses do not differ significantly in terms of host temperatures, because their hosts are either mammals or birds (Crawshaw, 1980), indicating that the observed enrichment of generalists' secretomes

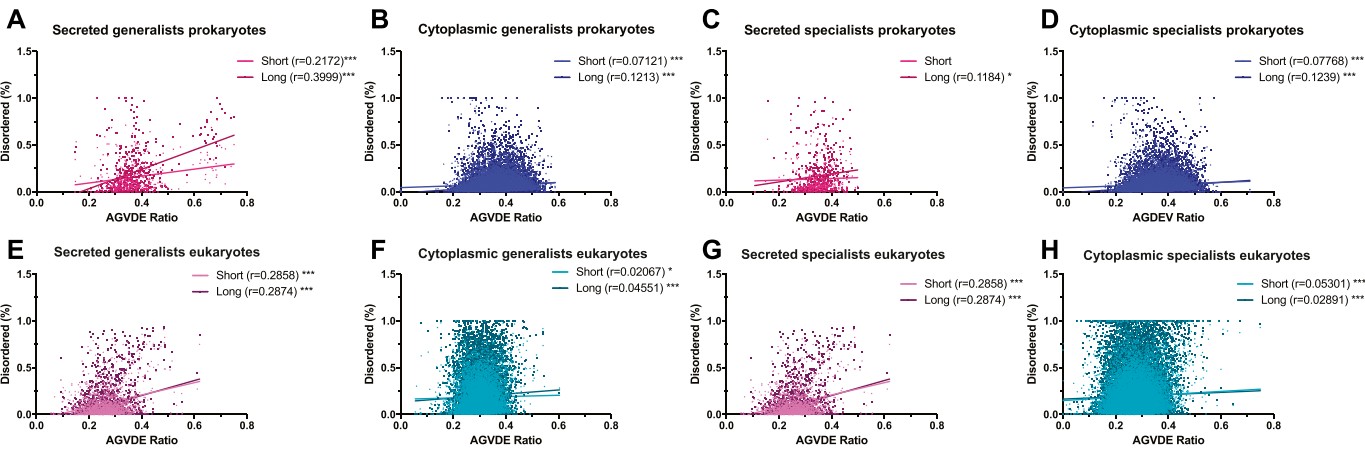

**Figure 3. Significant correlation between AGVDE ratio and intrinsic disorder in secreted proteins from generalists.**
The correlation between AGVDE ratio and the percentage of short or long stretches of disordered residues was determined in generalist (A, B, E, and F) and specialist (C, D, G, and H) pathogens. In (A–D) for 13 pairs of prokaryotic pathogens and in (E–H) for 3 pairs of eukaryotic pathogens, either in secreted (A, C, E, and G), and cytoplasmic (B, D, E, and H) derived proteins as indicated in Table 1. Shown are box-whiskers plots in which the line in the middle of the box represents the median value and whiskers are drawn down to the 10th percentile and up to the 90th percentile. Pearson's correlation coefficients (r) are shown in each graph. ***$P$ < 0.001; **$P$ < 0.01.

in primitive amino acids and intrinsically disordered regions are not due to differential temperature adaptation.

Other factors such as translation elongation speed and amino acid metabolic cost affect amino acid composition (Smith & Chapman, 2010; Gingold & Pilpel, 2011). It is thus conceivable that the observed differences in amino acid composition might be due to these factors differing between generalists and specialists. However, to our knowledge no such differences between generalists and specialists have been described, nor are there reasons to expect them. In addition, even if such differences existed, they would not be expected to distinctively affect secreted proteins.

Correlations between primitive amino acid content and disordered stretch content in prokaryotic generalist secreted proteins are probably so pronounced because primitive amino acids are disorder-promoting because of their lacking bulky and charged structures that constrain protein flexibility (Uversky, 2016). In agreement with these results, the disorder content of plant pathogen proteins has been recently characterized, showing that secreted proteins that are annotated as effectors are significantly enriched in intrinsic disorder, particularly in long disordered stretches (Marín & Ott, 2014). These authors proposed that the protein flexibility of the effectors is required to be transported through the type III secretion system (Marín & Ott, 2014).

Interestingly, high secretome sizes have been associated with zoonotic generalists compared with specialists (McNally et al, 2014). Our work further indicates that secretome composition is an important factor affecting host range. Furthermore, well-known are the functions of certain bacterial type I, type II, type III, type IV, type V, type VI, and type VII secreted proteins in host–pathogen interactions facilitating and promoting infection, acting as effectors targeting diverse specific host functions, or as powerful toxins disrupting host cellular homeostasis (Tseng et al, 2009), highlighting how relevant secreted proteins per se are in pathogenesis mechanisms. Our observations strongly suggest that pathogen secreted proteins play a role in influencing host range, which is also supported by the fact that the presence/absence of these secretion

systems and of certain effectors in *Salmonella* and *Pseudomonas syringae*, respectively, correlates with the range of colonization of different hosts (Baltrus et al, 2012; Pezoa et al, 2014).

In summary, the content of primitive amino acids in secreted proteins influences the host range of pathogens and this finding may be of global relevance and applicable to a broad variety of pathogens. Presently, the molecular determinants of species specificity remain poorly understood, probably because of the paradigm that membrane receptors are the main culprits in host–pathogen interactions; however, in this work, we show that the composition of membrane proteins is not strongly different between generalists and specialists. Adding new players, namely secreted proteins to our understanding of these interactions will facilitate gaining new insights and developing specific tools to combat infections. In this work, we describe the enrichment for disordered and primitive amino acids in the secretomes of broad-range pathogens compared with those of species-specific pathogens. The observations described here highlight the function of secreted proteins as key players in host–pathogen interactions.

## Materials and Methods

Protein and coding sequences were obtained from GenBank (https://www.ncbi.nlm.nih.gov/genbank/) and processed using in-house scripts. Protein subcellular locations were extracted from PSORTdb (prokaryotes) (Powell et al, 2016), MetazSecKB (metazoans) (Meinken et al, 2015), FunSecKB (*Candida*) (Lum & Min, 2011), and ProtSecKB (protists) (Peabody et al, 2016). Intrinsic disorder analyses were conducted using IUPred version 1.0 (Dosztanyi et al, 2005a,b), using a cutoff of 0.4 to classify disordered residues. For each membrane protein, its intracellular, transmembrane, and extracellular domains were identified using the TMHMM server, version 2 (http://www.cbs.dtu.dk/services/TMHMM/). Statistical analyses were conducted using the GraphPad Prism software (GraphPad Software) and the R package (R Core Team, 2013). *P*-values were

corrected for multiple testing using the Benjamini–Hochberg approach (Benjamini & Hochberg, 1995), and the corrected values are displayed in Tables S1–S6 in the Supplementary Information. The host ranges for most of the studied pathogens were retrieved from the Risk Group Database of the American Biological Safety Association (https://my.absa.org/tiki-index.php?page=Riskgroups) or from the sources listed in Table 1.

## Supplementary Information

## Acknowledgements

BL Payne, F Feyertag, and D Alvarez-Ponce were supported by funds from the University of Nevada, Reno, and by pilot grants from Nevada Institutional Developmental Award [IDea] Networks of Biomedical Research Excellence (P20GM103440) and the Smooth Muscle Plasticity Centers of Biomedical Research Excellence from the University of Nevada, Reno (5P30GM110767-04), both funded by the National Institute of General Medical Sciences (National Institutes of Health).

### Author Contributions

LP Blanco: conceptualization, data curation, formal analysis, supervision, validation, investigation, visualization, methodology, project administration, and writing—original draft, review, and editing.
B Payne: data curation, formal analysis, and methodology.
F Feyertag: data curation, formal analysis, and methodology.
D Alvarez-Ponce: conceptualization, resources, formal analysis, supervision, funding acquisition, validation, investigation, methodology, project administration, and writing—original draft, review, and editing.

### Conflict of Interest Statement

The authors declare that they have no conflict of interest.

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
