## [Reviewer comments · Life Science Alliance]

Proteins of generalist and specialist pathogens differ in their amino acid composition

Luz P. Blanco, Bryan L. Payne, Felix Feyertag, David Alvarez-Ponce
DOI: 10.26508/lsa.201800017

Review timeline:

Submission Date:	5 January 2018
Editorial Decision:	15 February 2018
Revision Received:	13 June 2018
Editorial Decision:	3 July 2018
Accepted:	10 July 2018

Report:

(Note: Letters and reports are not edited. The original formatting of letters and referee reports may not be reflected in this compilation.)

No Peer Review Process File is available with this article, as the authors have chosen not to make the review process public in this case.

1st Editorial Decision

15 February 2018

Thank you for submitting your manuscript entitled "Amino acid composition of secreted proteins defines the host range of pathogens" to Life Science Alliance.

Your manuscript was assessed by expert reviewers, whose comments are appended to this letter. The reviewers appreciate your work, but both think that your conclusions are not sufficiently supported by the data provided.

As it was not clear to me whether satisfactorily addressing the reviewer concerns is feasible, I invited you to comment on them prior to taking an editorial decision. I appreciate that you provided a preliminary point-by-point response to me upfront to outline how you would address the concerns raised by the reviewers. Based on your response, I would like to invite you to submit a revision, addressing the issues raised as you outlined. I should stress, however, that the outcome of the revision seems unclear at this stage and that a final positive recommendation will largely depend on your conclusions still holding true after adding the requested controls and quantifications.

Thank you for this interesting contribution to Life Science Alliance. We are looking forward to receiving your revised manuscript.

2nd Editorial Decision

3 July 2018

Thank you for submitting your revised manuscript entitled "Proteins of generalist and specialist pathogens differ in their amino acid composition". Your manuscript was assessed by the original reviewers again, whose reports are appended below.

As you will see, reviewer #2 appreciates the revision, while reviewer #1 points out that some aspects need further clarification. We would thus like to invite you to provide a final revised version of your work to address these remaining concerns. Importantly, please include the false discovery rate as initially outlined in your point-by-point response. Please also make sure to include all call-outs for the individual panels in Figure 3. Your response to reviewer #1, point 2 is sufficient in our view and does not need further revision. Once we receive the further revised version, we should be able to swiftly proceed to acceptance and publication of your work.
